# Direct Diffusion Bridge using Data Consistency for Inverse Problems

**Hyungjin Chung**[1]     **Jeongsol Kim**[1]     **Jong Chul Ye**[2]
[1] Dept. of Bio and Brain Engineering
[2]Graduate School of AI
Korea Advanced Institute of Science and Technology (KAIST)
`{hj.chung, jeongsol, jong.ye}@kaist.ac.kr`

## Abstract

Diffusion model-based inverse problem solvers have shown impressive performance, but are limited in speed, mostly as they require reverse diffusion sampling starting from noise. Several recent works have tried to alleviate this problem by building a diffusion process, directly bridging the clean and the corrupted for specific inverse problems. In this paper, we first unify these existing works under the name Direct Diffusion Bridges (DDB), showing that while motivated by different theories, the resulting algorithms only differ in the choice of parameters. Then, we highlight a critical limitation of the current DDB framework, namely that it does not ensure data consistency. To address this problem, we propose a modified inference procedure that imposes data consistency without the need for fine-tuning. We term the resulting method data Consistent DDB (CDDB), which outperforms its inconsistent counterpart in terms of both perception and distortion metrics, thereby effectively pushing the Pareto-frontier toward the optimum. Our proposed method achieves state-of-the-art results on both evaluation criteria, showcasing its superiority over existing methods. Code is open-sourced at `https://github.com/HJ-harry/CDDB`

## 1 Introduction

Diffusion models [15, 38] have become the de facto standard of recent vision foundation models [32, 31, 33]. Among their capabilities is the use of diffusion models as generative priors that can serve as plug-and-play building blocks for solving inverse problems in imaging [18, 38, 21, 5]. Diffusion model-based inverse problem solvers (DIS) have shown remarkable performance and versatility, as one can leverage the powerful generative prior regardless of the given problem at hand, scaling to linear [18, 38, 21], non-linear [5, 36], and noisy problems [21, 5].

Although there are many advantages of DIS, one natural limitation is its slow inference. Namely, the overall process of inference—starting from Gaussian noise and being repeatedly denoised to form a clean image—is kept the same, although there are marginal changes made to keep the sampling process consistent with respect to the given measurement. In such cases, the distance between the reference Gaussian distribution and the data distribution remains large, requiring inevitably a large number of sampling steps to achieve superior sample quality. On the other hand, the distribution of the measurements is much more closely related to the distribution of the clean images. Thus, intuitively, it would cost us much less compute if we were allowed to start the sampling process directly from the measurement, as in the usual method of direct inversion in supervised learning schemes.

Interestingly, several recent works aimed to tackle this problem under several different theoretical motivations: 1) Schrödinger bridge with paired data [26], 2) a new formulation of the diffusion process

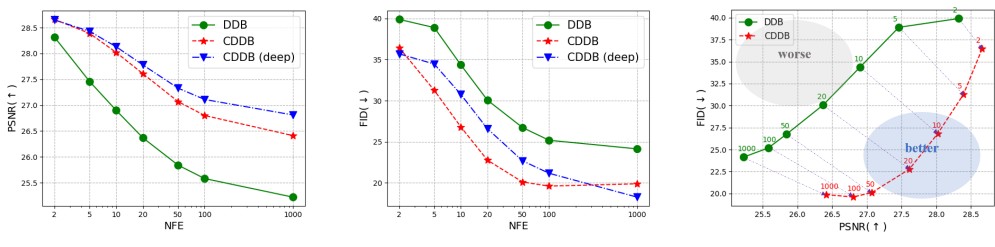

(a) NFE vs. PSNR (**higher** is better)   (b) NFE vs. FID (**lower** is better)   (c) Shift in pareto-frontier

Figure 1: Quantitative metric of I$^2$SB [26] (denoted DDB) vs. proposed CDDB on sr4x-bicubic task.

via constant-speed continual degradation [9], and 3) Ornstein-Uhlenbeck stochastic differential equation (OU-SDE) [28]. While developed from distinct motivations, the resulting algorithms can be understood in a unifying framework with minor variations: we define this new class of methods as **D**irect **D**iffusion **B**ridges (DDB; Section 3.1). In essence, DDB defines the diffusion process from the clean image distribution in $t = 0$ to the measurement distribution in $t = 1$ as the convex combination between the paired data, such that the samples $\boldsymbol{x}_t$ goes through continual degradation as $t = 0 \rightarrow 1$. In training, one trains a time-conditional neural network $G_\theta$ that learns a mapping to $\boldsymbol{x}_0$ for all timesteps, resulting in an iterative sampling procedure that *reverts* the measurement process.

Using such an iterative sampling process, one can flexibly choose the number of neural function evaluations (NFE) to generate reconstructions that meet the desiderata: with low NFE, less distortion can be achieved as the reconstruction regresses towards the mean [9]; with high NFE, one can opt for high perceptual quality at the expense of some distortion from the ground truth. This intriguing property of DDB creates a Pareto-frontier of reconstruction quality, where our desire would be to maximally pull the plot towards high perception and low distortion (bottom right corner of Fig. 1c).

In this work, we assert that DDB is missing a crucial component of *data consistency*, and devise methods to make the models *consistent* with respect to the given measurement by only modifying the sampling algorithm, without any fine-tuning of the pre-trained model. We refer to this new class of models as data **C**onsistent **D**irect **D**iffusion **B**ridge (CDDB; Section 3.2), and show that CDDB is capable of pushing the Pareto-frontier further towards the optima (lower distortion: Fig. 1a, higher perception: Fig. 1b, overall trend: Fig. 1c) across a variety of tasks. Theoretically, we show that CDDB is a generalization of DDS (Decomposed Diffusion Sampling) [6], a recently proposed method tailored for DIS with Gaussian diffusion, which guarantees stable and fast sampling. We then propose another variation, CDDB-deep, which can be derived as the DDB analogue of the DPS [5] by considering *deeper* gradients, which even further boosts the performance for certain tasks and enables the application to nonlinear problems where one cannot compute gradients in the usual manner (e.g. JPEG restoration). In the experiments, we showcase the strengths of each algorithm and show how one can flexibly construct and leverage the algorithms depending on the circumstances.

## 2   Background

### 2.1   Diffusion models

Diffusion models [15, 38, 22, 19] defines the forward data noising process $p(\boldsymbol{x}_t|\boldsymbol{x}_0)$ as

$$\boldsymbol{x}_t = \alpha_t \boldsymbol{x}_0 + \sigma_t \boldsymbol{z}, \; \boldsymbol{z} \sim \mathcal{N}(0, \boldsymbol{I}) \text{ for } t \in [0, 1], \tag{1}$$

where $\alpha_t, \sigma_t$ controls the signal component and the noise component, respectively, and are usually designed such that $\alpha_t^2 + \sigma_t^2 = 1$ [15, 22]. Starting from the data distribution $p_{\text{data}} := p(\boldsymbol{x}_0)$, the noising process in (1) gradually maps $p(\boldsymbol{x}_t)$ towards isotropic Gaussian distribution as $t \rightarrow 1$, i.e. $p(\boldsymbol{x}_1) \simeq \mathcal{N}(0, \boldsymbol{I})$. Training a neural network to *reverse* the process amounts to training a residual denoiser

$$\min_\theta \mathbb{E}_{\boldsymbol{x}_t \sim p(\boldsymbol{x}_t|\boldsymbol{x}_0), \boldsymbol{x}_0 \sim p_{\text{data}}(\boldsymbol{x}_0), \boldsymbol{\epsilon} \sim \mathcal{N}(0, \boldsymbol{I})} \left[ \| \boldsymbol{\epsilon}_\theta^{(t)}(\boldsymbol{x}_t) - \boldsymbol{\epsilon} \|_2^2 \right], \tag{2}$$

such that $\boldsymbol{\epsilon}_{\theta^*}^{(t)}(\boldsymbol{x}_t) \simeq \frac{\boldsymbol{x}_t - \alpha_t \boldsymbol{x}_0}{\sigma_t}$. Furthermore, it can be shown that epsilon matching is equivalent to the denoising score matching (DSM) [16, 37] objective up to a constant with different parameterization

$$\min_\theta \mathbb{E}_{\boldsymbol{x}_t, \boldsymbol{x}_0, \boldsymbol{\epsilon}} \left[ \| \boldsymbol{s}_\theta^{(t)}(\boldsymbol{x}_t) - \nabla_{\boldsymbol{x}_t} \log p(\boldsymbol{x}_t | \boldsymbol{x}_0) \|_2^2 \right], \tag{3}$$

such that $\boldsymbol{s}_{\theta^*}^{(t)}(\boldsymbol{x}_t) \simeq -\frac{\boldsymbol{x}_t - \alpha_t \boldsymbol{x}_0}{\sigma_t^2} = -\boldsymbol{\epsilon}_{\theta^*}^{(t)}(\boldsymbol{x}_t)/\sigma_t$. Moreover, for optimal $\theta^*$ and under regularity conditions, $\boldsymbol{s}_{\theta^*}(\boldsymbol{x}_t) = \nabla_{\boldsymbol{x}_t} \log p(\boldsymbol{x}_t)$. Then, sampling from the distribution can be performed by solving the reverse-time generative SDE/ODE [38, 19] governed by the score function. It is also worth mentioning that the posterior mean, or the so-called denoised estimate can be computed via Tweedie's formula [13]

$$\hat{\boldsymbol{x}}_{0|t} := \mathbb{E}_{p(\boldsymbol{x}_0 | \boldsymbol{x}_t)}[\boldsymbol{x}_0 | \boldsymbol{x}_t] = \frac{1}{\alpha_t}(\boldsymbol{x}_t + \sigma_t^2 \nabla_{\boldsymbol{x}_t} \log p(\boldsymbol{x}_t)) \simeq \frac{1}{\alpha_t}(\boldsymbol{x}_t + \sigma_t^2 \boldsymbol{s}_{\theta^*}^{(t)}(\boldsymbol{x}_t)). \tag{4}$$

In practice, DDPM/DDIM solvers [15, 35] work by iteratively refining these denoised estimates.

## 2.2 Diffusion model-based inverse problem solving with gradient guidance

Suppose now that we are given a measurement $\boldsymbol{y}$ obtained through some Gaussian linear measurement process $\boldsymbol{A}$, where our goal is to sample from the posterior distribution $p(\boldsymbol{x}|\boldsymbol{y})$. Starting from the sampling process of running the reverse SDE/ODE to sample from the prior distribution, one can modify the score function to adapt it for posterior sampling [38, 5]. By Bayes rule, $\nabla_{\boldsymbol{x}_t} \log p(\boldsymbol{x}_t|\boldsymbol{y}) = \nabla_{\boldsymbol{x}_t} \log p(\boldsymbol{x}_t) + \nabla_{\boldsymbol{x}_t} \log p(\boldsymbol{y}|\boldsymbol{x}_t)$, where $\nabla_{\boldsymbol{x}_t} \log p(\boldsymbol{x}_t) \simeq \boldsymbol{s}_{\theta^*}(\boldsymbol{x}_t)$. However, $\nabla_{\boldsymbol{x}_t} \log p(\boldsymbol{y}|\boldsymbol{x}_t)$ is intractable. Several methods have been proposed to approximate this time-dependent likelihood, two of the most widely used being DPS [5] and ΠGDM [36]. DPS proposes the following Jensen approximation[1]

$$\nabla_{\boldsymbol{x}_t} \log p(\boldsymbol{y}|\boldsymbol{x}_t) \overset{\text{(DPS)}}{\simeq} \nabla_{\boldsymbol{x}_t} \log p(\boldsymbol{y}|\hat{\boldsymbol{x}}_{0|t}) = \frac{\partial \hat{\boldsymbol{x}}_{0|t}}{\partial \boldsymbol{x}_t} \frac{\partial \|\boldsymbol{A}\hat{\boldsymbol{x}}_{0|t} - \boldsymbol{y}\|_2^2}{\partial \hat{\boldsymbol{x}}_{0|t}} = \underbrace{\frac{\partial \hat{\boldsymbol{x}}_{0|t}}{\partial \boldsymbol{x}_t}}_{\text{J}} \underbrace{\boldsymbol{A}^\top(\boldsymbol{y} - \boldsymbol{A}\hat{\boldsymbol{x}}_{0|t})}_{\text{V}}, \tag{5}$$

of which the chain rule is based on the denominator layout notation [41]. Here, we see that the gradient term can be represented as the Jacobian (J) vector (V) product (JVP). In the original implementation of DPS, the two terms are not computed separately, but computed directly as $\nabla_{\boldsymbol{x}_t} \|\boldsymbol{y} - \boldsymbol{A}\hat{\boldsymbol{x}}_{0|t}\|_2^2$, where the whole term can be handled with backpropagation. By this choice, DPS can also handle non-linear operators when the gradients can be computed, e.g. phase retrieval, forward model given as a neural network. On the other hand, ΠGDM proposes

$$\nabla_{\boldsymbol{x}_t} \log p(\boldsymbol{y}|\boldsymbol{x}_t) \overset{\text{(ΠGDM)}}{\simeq} \mathcal{N}(\boldsymbol{A}\hat{\boldsymbol{x}}_{0|t}, \boldsymbol{A}\boldsymbol{A}^\top + \boldsymbol{I}) = \underbrace{\frac{\partial \hat{\boldsymbol{x}}_{0|t}}{\partial \boldsymbol{x}_t}}_{\text{J}} \underbrace{\boldsymbol{A}^\dagger(\boldsymbol{y} - \boldsymbol{A}\hat{\boldsymbol{x}}_{0|t})}_{\text{V}}, \tag{6}$$

where $\boldsymbol{A}^\dagger := \boldsymbol{A}^\top (\boldsymbol{A}\boldsymbol{A}^\top)^{-1}$ is the Moore-Penrose pseudo-inverse. Using the JVP for implementation, it is no longer required that the whole term is differentiable. For this reason, ΠGDM can be applied to cases where we have non-differentiable, non-linear measurements given that an operation analogous to pseudo-inverse can be derived, e.g. JPEG restoration. Notably, the update step of DPS can be achieved by simply pre-conditioning ΠGDM with $\boldsymbol{A}^\top \boldsymbol{A}$. Implementing DIS with DPS (5) or ΠGDM (6) amounts to augmenting the gradient descent steps in between the ancestral sampling iterations.

While these methods are effective and outperforms the prior projection-based approaches [38, 21], they also have several drawbacks. Namely, the incorporation of the U-Net Jacobian is slow, compute-heavy, and often unstable [12, 34]. For example, when applied to MRI reconstruction in medical imaging, DPS results in noisy reconstructions [6] possibly due to unstable incorporation of the Wirtinger derivatives [23], and ΠGDM is hard to use as it is non-trivial to compute $\boldsymbol{A}^\dagger$. In order to circumvent these issues, DDS [6] proposed to use numerical optimization (i.e. conjugate gradients; CG) in the clean image domain, bypassing the need to compute J. Consequently, DDS achieves fast and stable reconstructions for inverse problems in medical imaging.

|  | $\mathbf{I^2SB}$ [26] | | **InDI** [9] |
| --- | --- | --- | --- |
| **Definition** | | | |
| Motivation | Schrödinger bridge | | Small-step MMSE |
| Base process | $\beta_t = \begin{cases} \beta_{\min} + 2\beta_{\mathrm{d}}t, & t \in [0, 0.5) \\ 2\beta_{\max} - 2\beta_{\mathrm{d}}t, & t \in [0.5, 1.0] \end{cases}$ | | - |
| | $\gamma_t = \int_0^t \beta_\tau \, d\tau, \; \bar{\gamma}_t = \int_t^1 \beta_\tau \, d\tau$ | | - |
| | Linear symmetric | | Const |
| **Diffusion process** | | | |
| $\alpha_t$ | $\gamma_t^2/(\gamma_t^2 + \bar{\gamma}_t^2)$ | | $t$ |
| $\sigma_t^2$ | $\gamma_t^2 \bar{\gamma}_t^2/(\gamma_t^2 + \bar{\gamma}_t^2)$ | | $t^2 \epsilon_t^2$ |
| **Sampling** | | | |
| $\alpha_{s\mid t}^2$ | $\gamma_s^2/\gamma_t^2$ | | $s/t$ |
| $\sigma_{s\mid t}^2$ | $\frac{(\gamma_t^2 - \gamma_s^2)\gamma_s^2}{\gamma_t^2}$ | | $s^2(\epsilon_s^2 - \epsilon_t^2)$ |

Table 1: Comparison between different types of DDB. $\beta_{\mathrm{d}} := \beta_{\max} - \beta_{\min}$. Further details are given in Appendix B.

## 3 Main Contributions

### 3.1 Direct Diffusion Bridge

We consider the case where we can sample $\boldsymbol{x}_0 := \boldsymbol{x} \sim p(\boldsymbol{x})$, and $\boldsymbol{x}_1 := \boldsymbol{y} \sim p(\boldsymbol{y}|\boldsymbol{x})$[2], i.e. paired data for training. Adopting the formulation of $\mathrm{I^2SB}$ [26] we define the posterior of $\boldsymbol{x}_t$ to be the product of Gaussians $\mathcal{N}(\boldsymbol{x}_t; \boldsymbol{x}_0, \gamma_t^2)$ and $\mathcal{N}(\boldsymbol{x}_t; \boldsymbol{x}_1, \bar{\gamma}_t^2)$, such that

$$p(\boldsymbol{x}_t|\boldsymbol{x}_0, \boldsymbol{x}_1) = \mathcal{N}\left(\boldsymbol{x}_t; \frac{\bar{\gamma}_t^2}{\gamma_t^2 + \bar{\gamma}_t^2}\boldsymbol{x}_0 + \frac{\gamma_t^2}{\gamma_t^2 + \bar{\gamma}_t^2}\boldsymbol{x}_1, \frac{\gamma_t^2 \bar{\gamma}_t^2}{\gamma_t^2 + \bar{\gamma}_t^2}\boldsymbol{I}\right). \tag{7}$$

Note that the sampling of $\boldsymbol{x}_t$ from (7) can be done by the reparametrization trick

$$\boldsymbol{x}_t = (1 - \alpha_t)\boldsymbol{x}_0 + \alpha_t \boldsymbol{x}_1 + \sigma_t \boldsymbol{z}, \; \boldsymbol{z} \sim \mathcal{N}(0, \boldsymbol{I}), \tag{8}$$

where $\alpha_t := \frac{\gamma_t^2}{\gamma_t^2 + \bar{\gamma}_t^2}$, $\sigma_t^2 := \frac{\gamma_t^2 \bar{\gamma}_t^2}{\gamma_t^2 + \bar{\gamma}_t^2}$[3]. This diffusion bridge introduces a *continual* degradation process by taking a convex combination of $(\boldsymbol{x}_0, \boldsymbol{x}_1)$, starting from the clean image at $t = 0$ to maximal degradation at $t = 1$, with additional stochasticity induced by the noise component $\sigma_t$. Our goal is to train a time-dependent neural network that maps any $\boldsymbol{x}_t$ to $\boldsymbol{x}_0$ that recovers the clean image. The training objective for $\mathrm{I^2SB}$ [26] analogous to denoising score matching (DSM) [16] reads

$$\min_\theta \mathbb{E}_{\boldsymbol{y} \sim p(\boldsymbol{y}|\boldsymbol{x}), \, \boldsymbol{x} \sim p(\boldsymbol{x}), \, t \sim U(0,1)} \left[\|\boldsymbol{s}_\theta(\boldsymbol{x}_t) - \frac{\boldsymbol{x}_t - \boldsymbol{x}_0}{\gamma_t}\|_2^2\right], \tag{9}$$

which is also equivalent to training a residual network $G_\theta$ with $\min_\theta \mathbb{E}[\|G_\theta(\boldsymbol{x}_t) - \boldsymbol{x}_0\|_2^2]$. For brevity, we simply denote the trained networks as $G_{\theta*}$ even if it is parametrized otherwise. Once the network is trained, we can reconstruct $\boldsymbol{x}_0$ starting from $\boldsymbol{x}_1$ by, for example, using DDPM ancestral sampling [15], where the posterior for $s < t$ reads

$$p(\boldsymbol{x}_s|\boldsymbol{x}_0, \boldsymbol{x}_t) = \mathcal{N}(\boldsymbol{x}_s; (1 - \alpha_{s|t}^2)\boldsymbol{x}_0 + \alpha_{s|t}^2 \boldsymbol{x}_t, \sigma_{s|t}^2 \boldsymbol{I}), \tag{10}$$

with $\alpha_{s|t}^2 := \frac{\gamma_s^2}{\gamma_t^2}$, $\sigma_{s|t}^2 := \frac{(\gamma_t^2 - \gamma_s^2)\gamma_s^2}{\gamma_t^2}$. At inference, $\boldsymbol{x}_0$ is replaced with a neural network-estimated $\hat{\boldsymbol{x}}_{0|t}$ to yield $\boldsymbol{x}_s \sim p(\boldsymbol{x}_s|\hat{\boldsymbol{x}}_{0|t}, \boldsymbol{x}_t)$.

However, when the motivation is to introduce 1) a tractable training objective that learns to recover the clean image along the degradation trajectory, and 2) devise a sampling method to gradually revert the degradation process, we find that the choices made for the parameters in (8),(7) can be arbitrary, as long as the marginal can be retrieved in the sampling process (10). In Table 1, we summarize the

---

[1] We ignore scaling constants that are related to the measurement noise for simplicity. For implementation, this can be absorbed into the choice of step sizes.

[2] For cases where there is a dimensionality mismatch, we use $\boldsymbol{x}_1 = \boldsymbol{A}^\dagger \boldsymbol{y}$. We keep this notation for simplicity.

[3] Hereafter, we override the definition of signal, noise coefficients $\alpha_t, \sigma_t$ that was first defined in Eq. (1).

| **Algorithm 1** CDDB | **Algorithm 2** CDDB (deep) |
|---|---|
| **Require:** $G_{\theta*}, \boldsymbol{x}_1, \alpha_i, \sigma_i, \alpha^2_{i-1|i}, \sigma^2_{i-1|i}, \rho_i$ | **Require:** $G_{\theta*}, \boldsymbol{x}_1, \alpha_i, \sigma_i, \alpha^2_{i-1|i}, \sigma^2_{i-1|i}, \rho_i$ |
| 1: **for** $i = N - 1$ to 0 **do** | 1: **for** $i = N - 1$ to 0 **do** |
| 2:   $\hat{\boldsymbol{x}}_{0|i} \leftarrow G_{\theta*}(\boldsymbol{x}_i)$ | 2:   $\hat{\boldsymbol{x}}_{0|i} \leftarrow G_{\theta*}(\boldsymbol{x}_i)$ |
| 3:   $\boldsymbol{z} \sim \mathcal{N}(\boldsymbol{0}, \boldsymbol{I})$ | 3:   $\boldsymbol{z} \sim \mathcal{N}(\boldsymbol{0}, \boldsymbol{I})$ |
| 4:   $\boldsymbol{x}'_{i-1} \leftarrow (1 - \alpha^2_{i-1|i})\hat{\boldsymbol{x}}_{0|i}$ $\quad + \alpha^2_{i-1|i}\boldsymbol{x}_i + \sigma_{i-1|i}\boldsymbol{z}$ | 4:   $\boldsymbol{x}'_{i-1} \leftarrow (1 - \alpha^2_{i-1|i})\hat{\boldsymbol{x}}_{0|i}$ $\quad + \alpha^2_{i-1|i}\boldsymbol{x}_i + \sigma_{i-1|i}\boldsymbol{z}$ |
| 5:   $\boldsymbol{g} \leftarrow \boldsymbol{A}^\top(\boldsymbol{y} - \boldsymbol{A}\hat{\boldsymbol{x}}_{0|i})$ | 5:   $\boldsymbol{g} \leftarrow \frac{\partial \hat{\boldsymbol{x}}_{0|i}}{\partial \boldsymbol{x}_i}\boldsymbol{A}^\dagger(\boldsymbol{y} - \boldsymbol{A}\hat{\boldsymbol{x}}_{0|i})$ |
| 6:   $\boldsymbol{x}_{i-1} \leftarrow \boldsymbol{x}'_{i-1} + \rho_{i-1}\boldsymbol{g}$ | 6:   $\boldsymbol{x}_{i-1} \leftarrow \boldsymbol{x}'_{i-1} + \rho_{i-1}\boldsymbol{g}$ |
| 7: **end for** | 7: **end for** |
| 8: **return** $\boldsymbol{x}_0$ | 8: **return** $\boldsymbol{x}_0$ |

choices made in [9, 26] to emphasize that the difference stems mostly from the parameter choices and not something fundamental. Concretely, sampling $\boldsymbol{x}_t$ from paired data can always be represented as (8): a convex combination of $\boldsymbol{x}_0$ and $\boldsymbol{x}_1$ with some additional noise. Reverse diffusion at inference can be represented as (10): a convex combination of $\boldsymbol{x}_0$ and $\boldsymbol{x}_t$ with some stochasticity. We define the methods that belong to this category as Direct Diffusion Bridge (DDB) henceforth. Below, we formally state the equivalence between the algorithms, with proofs given in Appendix A.

**Theorem 1.** *Let the parameters of InDI [9] in Table 1 be $t := \frac{\gamma_t^2}{\gamma_t^2 + \bar{\gamma}_t^2}$, $\epsilon_t^2 := \frac{\bar{\gamma}_t^2}{\gamma_t^2}(\gamma_t^2 + \bar{\gamma}_t^2)$. Then, InDI and $I^2SB$ are equivalent.*

The equivalence relation will be useful when we derive our CDDB algorithm in Section. 3.2. As a final note, IR-SDE [28] does not strictly fall into this category as the sampling process is derived from running the reverse SDE. However, the diffusion process can still be represented as (8) by setting $\alpha_t = 1 - e^{-\bar{\theta}_t}$, $\sigma_t^2 = \lambda^2(1 - e^{-2\bar{\theta}_t})$, and the only difference comes from the sampling procedure.

### 3.2   Data Consistent Direct Diffusion Bridge

**Motivation**   Regardless of the choice in constructing DDB, there is a crucial component that is missing from the framework. While the sampling process (10) starts directly from the measurement (or equivalent), as the predictions $\hat{\boldsymbol{x}}_{0|t} = G_\theta(\boldsymbol{x}_t)$ are imperfect and are never guaranteed to preserve the measurement condition $\boldsymbol{y} = \boldsymbol{Ax}$, the trajectory can easily deviate from the desired path, while the residual blows up. Consequently, this may result in inferior sample quality, especially in terms of distortion. In order to mitigate this downside, our strategy is to keep the DDB sampling strategy (10) intact and augment the steps to constantly *guide* the trajectory to satisfy the data consistency, similar in spirit to gradient guidance in DIS. Here, we focus on the fact that the clean image estimates $\hat{\boldsymbol{x}}_{0|t}$ is produced at every iteration, which can be used to compute the residual with respect to the measurement $\boldsymbol{y}$. Taking a gradient step that minimizes this residual after every sampling step results in Algorithm 1, which we name data Consistent DDB (CDDB). In the following, we elaborate on how the proposed method generalizes DDS which was developed for DIS.

**CDDB as a generalization of DDS [6]**   Rewriting (10) with reparameterization trick

$$\boldsymbol{x}_s = \underbrace{\hat{\boldsymbol{x}}_{0|t}}_{\text{Denoise}(\boldsymbol{x}_t)} + \underbrace{\alpha^2_{s|t}(\boldsymbol{x}_t - \hat{\boldsymbol{x}}_{0|t})}_{\text{Noise}(\boldsymbol{x}_t)} + \sigma_{s|t}\boldsymbol{z}, \quad \hat{\boldsymbol{x}}_{0|t} := G_{\theta*}(\boldsymbol{x}_t) \tag{11}$$

we see that the iteration decomposes into three terms: the denoised component, the deterministic noise, and the stochastic noise. The key observation of DDIM [35] is that if the score network is fully expressive, then the deterministic noise term $\boldsymbol{x}_t - \hat{\boldsymbol{x}}_{0|t}$ becomes Gaussian such that it satisfies the total variance condition

$$\left(\alpha^2_{s|t}\sigma_t\right)^2 + \sigma^2_{s|t} = \sigma^2_s, \tag{12}$$

allowing (11) to restore the correct marginal $\mathcal{N}(\boldsymbol{x}_s; \boldsymbol{x}_0, \sigma_s^2)$. Under this condition, DDS showed that using a few step of numerical optimization ensure the updates from the denoised image $\hat{\boldsymbol{x}}_{0|t}$ remain on the clean manifold. Furthermore, subsequent noising process using deterministic and stochastic noises can then be used to ensure the transition to the correct noisy manifold [6].

Under this view, our algorithm can be written concisely as

$$\boldsymbol{x}_s \leftarrow \underbrace{\hat{\boldsymbol{x}}_{0|t} + \rho \boldsymbol{A}^\top (\boldsymbol{y} - \boldsymbol{A}\hat{\boldsymbol{x}}_{0|t})}_{\text{CDenoise}(\boldsymbol{x}_t)} + \underbrace{\alpha_{s|t}^2 (\boldsymbol{x}_t - \hat{\boldsymbol{x}}_{0|t})}_{\text{Noise}(\boldsymbol{x}_t)} + \sigma_{s|t}\boldsymbol{z}, \tag{13}$$

where we make the update step only to the clean denoised component, and leave the other components as is. In order to achieve proper sampling that obeys the marginals, it is important to show that the remaining components constitute the correct noise variance and the condition assuming Gaussianity should be (12). In the following, we show that this is indeed satisfied for the two cases of direction diffusion bridge (DDB):

**Theorem 2.** *The total variance condition* (12) *is satisfied for both I²SB and InDI.*

*Proof.* For InDI, considering the noise variance $\sigma_t = t\epsilon_t$ in Table 1,

$$\left(\alpha_{s|t}^2 \sigma_t\right)^2 + \sigma_{s|t}^2 = \frac{s^2}{t^2} t^2 \epsilon_t^2 + s^2(\epsilon_s^2 - \epsilon_t^2) = s^2 \epsilon_s^2 = \sigma_s^2. \tag{14}$$

Due to the equivalence in Theorem 1, the condition is automatically satisfied in I²SB. We show that this is indeed the case in Appendix A. □

In other words, given that the gradient descent update step in $\text{CDenoise}(\boldsymbol{x}_t)$ does not leave the clean data manifold, it is guaranteed that the intermediate samples generated by (13) will stay on the correct noisy manifold [6]. In this regard, CDDB can be thought of as the DDB-generalized version of DDS. Similar to DDS, CDDB does not require the computation of heavy U-Net Jacobians and hence introduces negligible computation cost to the inference procedure, while being robust in the choice of step size.

**CDDB-deep** As shown in DPS and ΠGDM, taking *deeper* gradients by considering U-Net Jacobians is often beneficial for reconstruction performance. Moreover, it even provides way to impose data consistency for non-linear inverse problems, where standard gradient methods are not feasible. In order to devise an analogous method, we take inspiration from DPS, and propose to augment the solver with a gradient step that maximizes the time-dependent likelihood (w.r.t. the measurement) $p(\boldsymbol{y}|\boldsymbol{x}_t)$. Specifically, we use the Jensen approximation from [5]

$$p(\boldsymbol{y}|\boldsymbol{x}_t) = \int p(\boldsymbol{y}|\boldsymbol{x}_0)p(\boldsymbol{x}_0|\boldsymbol{x}_t)\,d\boldsymbol{x}_0$$
$$= \mathbb{E}_{p(\boldsymbol{x}_0|\boldsymbol{x}_t)}[p(\boldsymbol{y}|\boldsymbol{x}_0)] \simeq p(\boldsymbol{y}|\mathbb{E}[\boldsymbol{x}_0|\boldsymbol{x}_t]) = p(\boldsymbol{y}|\hat{\boldsymbol{x}}_{0|t}), \tag{15}$$

where the last equality is naturally satisfied from the training objective (9). Using the approximation used in (15), the correcting step under the Gaussian measurement model yields

$$\nabla_{\boldsymbol{x}_t} \log p(\boldsymbol{y}|\boldsymbol{x}_t) \simeq \nabla_{\boldsymbol{x}_t} \|\boldsymbol{y} - \boldsymbol{A}\hat{\boldsymbol{x}}_{0|t}\|_2^2. \tag{16}$$

Implementing (16) in the place of the shallow gradient update step of Algorithm 1, we achieve CDDB-deep (see Algorithm 2). From our initial experiments, we find that preconditioning with $\boldsymbol{A}^\dagger$ as in ΠGDM improves performance by a small margin, and hence use this setting as default.

## 4 Experiments

### 4.1 Setup

**Model, Dataset** For a representative DDB, we choose I²SB [26] along with the pre-trained model weights for the following reasons: 1) it is open-sourced[4], 2) it stands as the current state-of-the-art, 3) the model architecture is based on ADM [11], which induces fair comparison against other DIS methods. All experiments are based on ImageNet 256×256 [10], a benchmark that is considered to be much more challenging for inverse problem solving based on generative models [5], compared to more focused datasets such as FFHQ [20]. We follow the standards of [26] and test our method on the following degradations: sr4x-{bicubic, pool}, deblur-{uniform, gauss}, and JPEG restoration with 1k validation images.

---

[4]https://github.com/NVlabs/I2SB

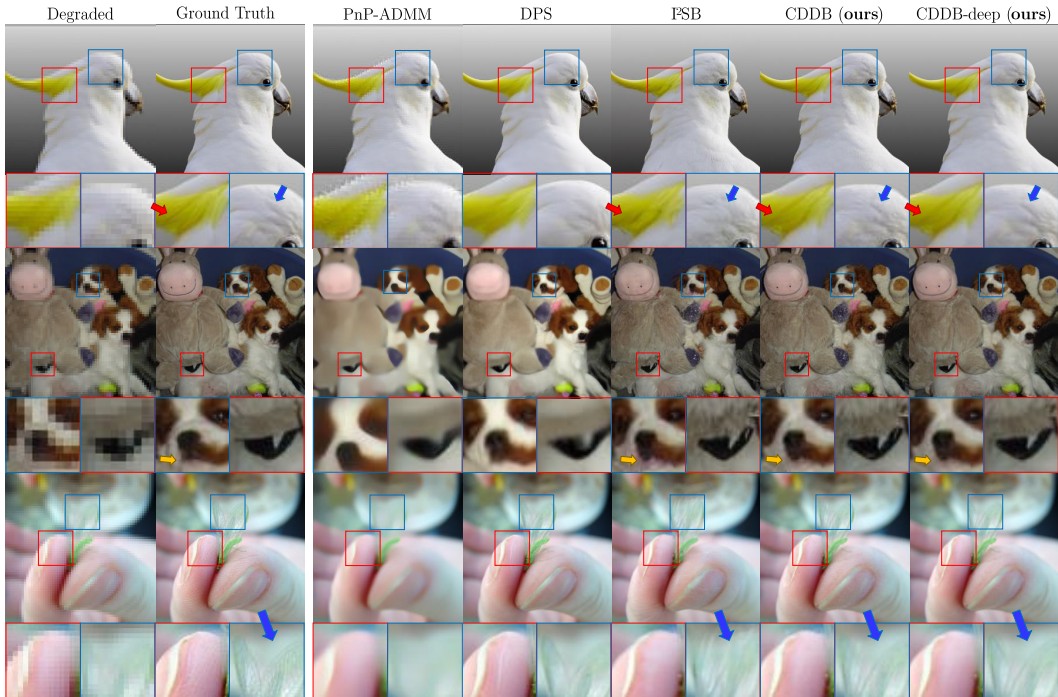

Figure 2: SR(×4)-bicubic reconstruction results. CDDB/CDDB-deep corrects details wrongly captured in I²SB: line texture in 1st row, color/texture errors in 2nd row, topology of wings in 3rd row (see arrows).

| | SR(×4) | | | | | | | | Deblur | | | | | | | |
|---|---|---|---|---|---|---|---|---|---|---|---|---|---|---|---|---|
| | bicubic | | | | pool | | | | gauss | | | | uniform | | | |
| Method | PSNR ↑ | SSIM ↑ | LPIPS ↓ | FID ↓ | PSNR ↑ | SSIM ↑ | LPIPS ↓ | FID ↓ | PSNR ↑ | SSIM ↑ | LPIPS ↓ | FID ↓ | PSNR ↑ | SSIM ↑ | LPIPS ↓ | FID ↓ |
| CDDB (ours) | **26.41** | **0.860** | **0.198** | **19.88** | 26.36 | **0.855** | **0.184** | **17.79** | **37.02** | **0.978** | **0.059** | 5.007 | **31.26** | **0.927** | **0.193** | 23.15 |
| I²SB [26] | 25.22 | 0.802 | 0.260 | 24.13 | 25.08 | 0.800 | 0.258 | 23.53 | 36.01 | 0.973 | 0.067 | 5.800 | 30.75 | 0.919 | 0.198 | 23.01 |
| DPS [5] | 19.89 | 0.498 | 0.384 | 63.37 | 21.01 | 0.562 | 0.326 | 49.34 | 27.21 | 0.766 | 0.244 | 34.58 | 22.51 | 0.565 | 0.357 | 60.00 |
| ΠGDM [36] | 26.20 | 0.850 | 0.252 | 29.36 | 26.07 | 0.849 | 0.256 | 26.97 | - | - | - | - | - | - | - | - |
| DDRM [21] | 26.05 | 0.838 | 0.270 | 46.49 | 25.54 | 0.848 | 0.257 | 40.40 | 36.73 | 0.975 | 0.071 | **4.346** | 29.21 | 0.901 | 0.210 | **19.97** |
| DDNM [40] | 26.41 | 0.801 | 0.230 | 38.63 | 26.04 | 0.792 | 0.218 | 33.15 | - | - | - | - | - | - | - | - |
| DDS [6] | 26.41 | 0.801 | 0.230 | 38.64 | 26.04 | 0.792 | 0.218 | 33.15 | 33.27 | 0.945 | 0.057 | 6.442 | 27.88 | 0.829 | 0.193 | 26.07 |
| PnP-ADMM [4] | 26.16 | 0.788 | 0.350 | 74.06 | 25.85 | 0.733 | 0.372 | 72.63 | 28.18 | 0.800 | 0.325 | 60.27 | 25.47 | 0.701 | 0.416 | 83.76 |
| ADMM-TV | 22.55 | 0.595 | 0.493 | 122.7 | 22.31 | 0.574 | 0.512 | 119.4 | 24.67 | 0.773 | 0.324 | 50.74 | 21.72 | 0.600 | 0.491 | 98.15 |

Table 2: Quantitative evaluation of SR, deblur task on ImageNet 256×256-1k. **Bold**: Best, under: second best. Colored: DDB methods.

**Baselines, Evaluation**   Along with the most important comparison against I²SB, we also include comparisons with state-of-the-art DIS methods including DDRM [21], DPS [5], ΠGDM [36], DDNM [40], and DDS [6]. For choosing the NFE and the hyper-parameters for each method, we closely abide to the original advised implementation: DDRM (20 NFE), DPS (1000 NFE), ΠGDM (100 NFE), DDNM (100 NFE), DDS (100 NFE). We find that increasing the NFE for methods other than DPS does not induce performance gain. We also note that we exclude ΠGDM and DDNM for the baseline comparison in the deblurring problem, as directly leveraging the pseudo-inverse matrix may result in unfair boost in performance [29]. We additionally perform comparisons against PnP-ADMM [4] and ADMM-TV, which are methods tailored towards higher SNR. For I²SB along with the proposed method, we choose 100 NFE for JPEG restoration as we found it to be the most stable, and choose 1000 NFE for all other tasks. Further details on the experimental setup can be found in Appendix C.

## 4.2   Results

**Comparison against baselines**   Throughout the experiments, we thoroughly analyze both distortion and perception of the reconstructions obtained through our method against other DDB, DIS, and iterative optimization methods. Note that for the recent diffusion-based methods, analysis has been

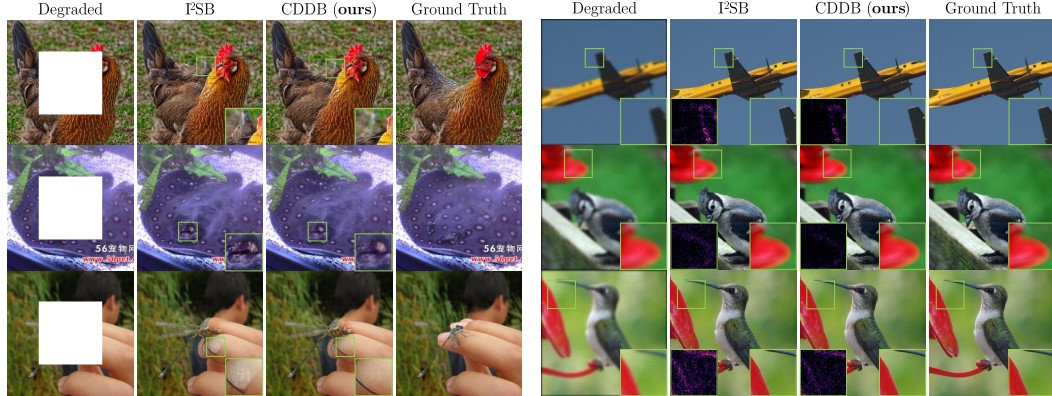

Figure 3: Results on inpainting (Left) and deblurring (Right). For inpainting, boundaries are corrected (row 1) and artifacts are corrected/removed (row 2,3). For deblurring, halo artifacts are corrected, and grid patterns from the background are alleviated.

mainly focused on perceptual metrics [5, 36, 26], mainly because these methods excel on these metrics, but often compromising distortion metrics. DDB methods often take this to the extreme, where one can achieve the best PSNR with very little NFE, and the PSNR consistently degrades as one increases the NFE [9, 26] (See Fig. 1c). Despite this fact, the standard setting in DDB methods is to set a high NFE as one can achieve much improved perceptual quality. On the other hand, conventional iterative methods are often highly optimized for less distortion, albeit with low perceptual quality. While this trade-off may seem imperative, we show that CDDB can improve both aspects, putting it in the place of the state-of-the-art on most experiments (See Tab. 2, 3). A similar trend can be observed in Fig. 2, where we see that CDDB greatly improves the performance of DDB, while also outperforming DIS and iterative optimization methods.

**CDDB pushes forward the Pareto-frontier** It is widely known that there exists an inevitable trade-off of distortion when aiming for higher perceptual quality [3, 24]. This phenomenon has been reconfirmed consistently in the recent DIS [5, 36] and DDS [9, 26] methods. For DDB, one can flexibly control this trade-off by simply choosing different NFE values, creating a Pareto-frontier with higher NFE tailored towards perceptual quality. While this property is intriguing, the gain we achieve when increasing the

| Method | PSNR ↑ | SSIM ↑ | LPIPS ↓ | FID ↓ |
|---|---|---|---|---|
| CDDB (ours) | **26.34** | 0.837 | **0.263** | **19.48** |
| I²SB [26] | 26.12 | 0.832 | 0.266 | 20.35 |
| ΠGDM [36] | 26.09 | **0.842** | 0.282 | 30.27 |
| DDRM [21] | 26.33 | 0.829 | 0.330 | 47.02 |

Table 3: Quantitative evaluation of the JPEG restoration (QF = 10) task.

NFE decreases *exponentially*, and eventually reaches a bound when NFE > 1000. In contrast, we show in Fig. 1 that CDDB pushes the bound further towards the optima. Specifically, 20 NFE CDDB *outperforms* 1000 NFE DDB in PSNR by > 2 db, while having lower FID (i.e. better perceptual quality). To this point, CDDB induces dramatic acceleration (> 50×) to DDB.

**CDDB vs. CDDB-deep** The two algorithms presented in this work share the same spirit but have different advantages. CDDB generally has higher speed and stability, possibly due to guaranteed convergence. As a result, it robustly increases the performance of SR and deblurring. In contrast, considering the case of inpainting and JPEG restoration, CDDB cannot improve the performance of DDB. For inpainting, the default setting of I²SB ensures consistency by iteratively applying replacement, as implemented in [40]. As the measurement stays in the pixel space, the gradients cannot impose any constraint on the missing pixel values. CDDB-deep is useful in such a situation, as the U-Net Jacobian has a global effect on *all* the pixels, improv-

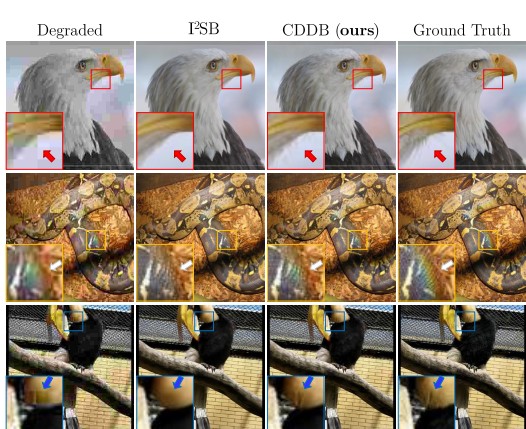

Figure 4: Results on JPEG restoration (QF=10). CDDB recovers texture details (row 1,3), and color details (row 2).

ing the performance by inducing coherence. CDDB-deep also enables the extension to nonlinear inverse problems where one cannot take standard gradient steps. This is illustrated for the case of JPEG restoration in Tab. 3 and Fig. 4, where we see overall improvement in performance compared to I$^2$SB.

**Noise robustness**  DIS methods are often designed such that they are robust to measurement noise [21, 5]. In contrast, this is not the case for DDB as they are trained in a supervised fashion: If not explicitly trained with synthetic adding of noise, the method does not generalize well to noisy measurements, as can be seen in Fig. 5. On the other hand, note that with CDDB, we are essentially incorporating a Gaussian likelihood model, which naturally enhances the robustness to noise. As a result, while I$^2$SB tends to propagate noise (best seen in the background), we do not observe such artifacts when using CDDB.

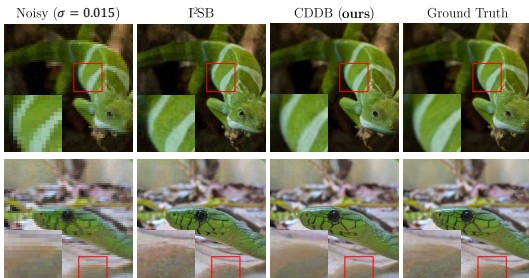

Figure 5: Results on noisy SR×4 reconstruction. I$^2$SB propagates noise to the reconstruction. CDDB effectively removes noise.

## 5    Discussion

**Extension to other related works**  Going beyond the paired inverse problem setting and considering the Schrödinger Bridge (SB) problem [25, 8], or more generally transport mapping problems [27] between the two unmatched distributions, it is often desirable to control the deviation from the start of sampling. A concrete example would be the case of image-to-image translation [43] where one does not want to alter the content of the image. As CDDB can be thought of as a regularization method that penalizes the deviation from the starting point, the application is general and can be extended to such SB problems at inference time by using the gradients that minimize the distance from the start point. We leave this direction for future work.

**Data consistency in supervised learning frameworks**  The first applications of supervised deep learning to solve inverse problems in medical imaging (e.g. CT [17], MRI reconstruction [39]) mostly involved directly inverting the measurement signal without considering the measurement constraints. The works that followed [1, 14] naturally extended the algorithms by incorporating measurement consistency steps in between the forward passes through the neural network. Analogously, CDDB is a natural extension of DDB but with high flexibility, as we do not have to pre-determine the number of forward passes [1] or modify the training algorithm [14].

## 6    Conclusion

In this work, we unify the seemingly different algorithms under the class of direct diffusion bridges (DDB) and identify the crucial missing part of the current methods: data consistency. Our train-free modified inference procedure named consistent DDB (CDDB) fixes this problem by incorporating consistency-imposing gradient steps in between the reverse diffusion steps, analogous to the recent DIS methods. We show that CDDB can be seen as a generalization of representative DIS methods (DDS, DPS) in the DDB framework. We validate the superiority of our method with extensive experiments on diverse inverse problems, achieving state-of-the-art sample quality in both distortion and perception. Consequently, we show that CDDB can push the Pareto-frontier of the reconstruction toward the desired optimum.

**Limitations and societal impact**  The proposed method assumes prior knowledge of the forward operator. While we limit our scope to non-blind inverse problems, the extension of CDDB to blind inverse problems [7, 30] will be a possible direction of research. Moreover, for certain inverse problems (e.g. inpainting), even when do observe improvements in qualitative results, the quantitative metrics tend to slightly decrease overall. Finally, inheriting from DDS/DIS methods, our method relies on strong priors that are learned from the training data distribution. This may potentially lead to reconstructions that intensify social bias and should be considered in practice.

## Acknowledgments and Disclosure of Funding

This research was supported by the KAIST Key Research Institute (Interdisciplinary Research Group) Project, by the National Research Foundation of Korea under Grant NRF-2020R1A2B5B03001980, by the Korea Medical Device Development Fund grant funded by the Korea government (the Ministry of Science and ICT, the Ministry of Trade, Industry and Energy, the Ministry of Health & Welfare, the Ministry of Food and Drug Safety) (Project Number: 1711137899, KMDF_PR_20200901_0015), by Institute of Information & communications Technology Planning & Evaluation (IITP) grant funded by the Korea government(MSIT) (No.2019-0-00075, Artificial Intelligence Graduate School Program(KAIST)), and by the Field-oriented Technology Development Project for Customs Administration through National Research Foundation of Korea(NRF) funded by the Ministry of Science & ICT and Korea Customs Service(NRF-2021M3I1A1097938).

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
