# A Proofs

**Theorem 1.** *Let the parameters of InDI [9] in Table 1 be $t := \frac{\gamma_t^2}{\gamma_t^2 + \bar{\gamma}_t^2}$, $\epsilon_t^2 := \frac{\bar{\gamma}_t^2}{\gamma_t^2}(\gamma_t^2 + \bar{\gamma}_t^2)$. Then, InDI and $I^2SB$ are equivalent.*

*Proof.* In order to proceed with the proof, we first remind the notations from Table 1:

$$\gamma_t^2 := \int_0^t \beta_\tau \, d\tau, \quad \bar{\gamma}_t^2 := \int_t^1 \beta_\tau \, d\tau. \tag{17}$$

By definition, we note that $\gamma_t^2 + \bar{\gamma}_t^2 = \int_0^1 \beta_\tau \, d\tau$ is constant for any choice of $t \in [0, 1]$. To proceed with the proof, we have for $\alpha_{s|t}^2$,

$$s/t = \frac{\gamma_s^2}{\gamma_s^2 + \bar{\gamma}_s^2} \bigg/ \frac{\gamma_t^2}{\gamma_t^2 + \bar{\gamma}_t^2} = \frac{\gamma_s^2}{\gamma_t^2}. \tag{18}$$

Considering $\sigma_{s|t}^2$,

$$s^2(\epsilon_s^2 - \epsilon_t^2) = \frac{\gamma_s^4}{(\gamma_s^2 + \bar{\gamma}_s^2)^2}\left(\frac{\bar{\gamma}_s^2}{\gamma_s^2}(\gamma_s^2 + \bar{\gamma}_s^2) - \frac{\bar{\gamma}_t^2}{\gamma_t^2}(\gamma_t^2 + \bar{\gamma}_t^2)\right) \tag{19}$$

$$= \frac{\gamma_s^4}{\gamma_s^2 + \bar{\gamma}_s^2}\left(\frac{\bar{\gamma}_s^2}{\gamma_s^2} - \frac{\bar{\gamma}_t^2}{\gamma_t^2}\right) \tag{20}$$

$$= \frac{\gamma_s^2}{\gamma_s^2 + \bar{\gamma}_s^2}\left(\frac{\bar{\gamma}_s^2\gamma_t^2 - \bar{\gamma}_t^2\gamma_s^2}{\gamma_t^2}\right) \tag{21}$$

$$\overset{(a)}{=} \frac{\gamma_s^2}{\gamma_s^2 + \bar{\gamma}_s^2}\frac{d_s^2(\gamma_t^2 + \bar{\gamma}_t^2)}{\gamma_t^2} \tag{22}$$

$$= \frac{\gamma_s^2}{\gamma_t^2}d_s^2 \tag{23}$$

$$= \frac{\gamma_s^2}{\gamma_t^2}(\gamma_t^2 - \gamma_s^2), \tag{24}$$

where in (a), we defined $d_s^2 := \int_s^t \beta_\tau, d\tau$ such that $\gamma_s^2 + d_s^2 = \gamma_t^2$ is satisfied. $\qquad\square$

**Theorem 2.** *The total variance condition (12) is satisfied for both $I^2SB$ and InDI.*

*Proof.* Here, we further show that the condition is satisfied for $I^2SB$ for completeness.

$$\sigma_s^2 - (\alpha_{s|t}^2)^2\sigma_t^2 = \frac{\gamma_s^2\bar{\gamma}_s^2}{\gamma_s^2 + \bar{\gamma}_s^2} - \frac{\gamma_s^4}{\gamma_t^4}\frac{\gamma_t^2\bar{\gamma}_t^2}{\gamma_t^2 + \bar{\gamma}_t^2} \tag{25}$$

$$= \frac{\gamma_s^2}{\gamma_s^2 + \bar{\gamma}_s^2}\left(\bar{\gamma}_s^2 - \frac{\gamma_s^2}{\gamma_t^2}\bar{\gamma}_t^2\right) \tag{26}$$

$$= \frac{\gamma_s^2}{\gamma_t^2}\frac{\bar{\gamma}_s^2\gamma_t^2 - \gamma_s^2\bar{\gamma}_t^2}{\gamma_s^2 + \bar{\gamma}_s^2} \tag{27}$$

$$= \frac{\gamma_s^2}{\gamma_t^2}\frac{(\bar{\gamma}_t^2 + d_s^2)\gamma_t^2 - \gamma_s^2(\gamma_t^2 - d_s^2)}{\gamma_s^2 + \bar{\gamma}_s^2} \tag{28}$$

$$= \frac{\gamma_s^2}{\gamma_t^2}\frac{d_s^2(\gamma_t^2 + \bar{\gamma}_t^2)}{\gamma_s^2 + \bar{\gamma}_s^2} \tag{29}$$

$$= \frac{\gamma_s^2(\gamma_t^2 - \gamma_s^2)}{\gamma_t^2} = \sigma_{s|t}^2, \tag{30}$$

where we use $\gamma_s^2 + d_s^2 = \gamma_t^2$ for the last equality. $\qquad\square$

# B  Details on Table 1

**I$^2$SB**  $\beta_{\min} = 0.0001$, $\beta_{\max} = 0.02$. The same $\beta_t$ schedule is used regardless of being implemented as deterministic or not. For the former, $\sigma_t = \sigma^2_{s|t} = 0$.

**InDI**  The speed of the base degradation process is parametrized with constant time-speed $t$. For deterministic methods, $\epsilon_t = 0$ and hence $\sigma_t = \sigma^2_{s|t} = 0$. For stochastic methods, $\epsilon_t = 0.01$ such that $\sigma^2_{s|t} = 0$.

**IR-SDE (PF-ODE)**  As mentioned in the main text, IR-SDE can also be considered a type of DDB as it obeys the marginal distribution that is characterized by (8), but with a different sampling method. Specifically, we can set $\alpha_t = 1 - e^{-\bar{\theta}_t}$, $\sigma^2_t = \lambda^2(1 - e^{-2\bar{\theta}_t})$, with $\delta = 0.008$. $\lambda^2$ is the stationary variance of the OU-process, which is set to the noise variance of the degraded image. For all non-noisy inverse problems that we mostly consider in this work, $\lambda^2$ is still set to $10/255$.

# C  Experimental Details

## C.1  Implementation Details

**Models**  All the models that are used throughout the work are based on pre-trained models from I$^2$SB[5]. All the models are fine-tuned from the ADM [11] ImageNet 256×256 model, hence share the same architecture as well as the specific model hyper-parameter settings.

**Inverse problem setting**  All the forward operators $\boldsymbol{A}$ are used in the same manner as in [26], which are adopted from DDRM [21].

**Step size, gradient**  For both Algorithms 1,2, we use constant step size, but scaled to match the signal ratio of the intermediate reconstructions $\hat{\boldsymbol{x}}_{0|i}$: $\rho_i = (1 - \alpha^2_{i|i+1})c$, where $c$ is some constant. For SR and deblurring tasks, we take $c = 1.0$. For JPEG restoration, we take $c = 0.5$.

When implementing the gradient computation for CDDB, we do not explicitly compute $\boldsymbol{A}^\top$, but rely on automatic differentiation for simplicity. This lets us unify the implementations of Algorithm 1,2 in a similar manner.

**Compute**  All experiments were run using a single RTX 3090 GPU. On average, I$^2$SB and CDDB with 1000 NFE take about 82 seconds ($\sim 0.08$ sec. / iter). CDDB-deep takes about 193 seconds ($\sim 0.19$ sec. / iter).

**Code Availability**  Code is available at `https://github.com/HJ-harry/CDDB`

## C.2  Comparison Methods

**I$^2$SB**  We follow the default setting advised in [26] and set the default sampling schedule to be the quadratic schedule [35] with 1000 NFE. As we leverage the pre-trained checkpoints provided, deblurring and inpainting models are set to OT-ODE with no additive Gaussian noise during the sampling process.

**DPS, ΠGDM**  Pre-trained ADM models that were used in the original work [5, 36] are used. For DPS, we use the constant step size of 1.0 with 1000 NFE. For ΠGDM , we use the constant step size of 1.0 with 100 NFE. We initially experimented with higher NFE for ΠGDM but did not find a boost in performance.

**DDRM**  We use 20 NFE DDIM sampling with $\eta = 0.85$, $\eta_b = 1.0$ as advised.

**DDNM**  We use 100 NFE sampling with $\eta = 0.85$ as advised. We do not use time travel for all experiments.

---

[5]https://github.com/NVlabs/I2SB

| $c =$ | CDDB | | | | | | | | CDDB-deep | | | | | | | |
|---|---|---|---|---|---|---|---|---|---|---|---|---|---|---|---|---|
| | 0.0 | 0.25 | 0.5 | 0.75 | 1.0 | 1.5 | 2.0 | 3.0 | 0.0 | 0.25 | 0.5 | 0.75 | 1.0 | 1.5 | 2.0 | 3.0 |
| PSNR (↑) | 25.07 | 26.14 | 26.28 | **26.31** | **26.31** | 26.30 | 26.24 | 25.70 | 25.07 | 26.21 | 26.42 | 26.51 | 26.56 | **26.59** | 26.57 | 26.06 |
| SSIM (↑) | 0.692 | 0.744 | 0.752 | **0.753** | **0.753** | 0.751 | 0.747 | 0.729 | 0.692 | 0.745 | 0.756 | 0.760 | 0.761 | **0.762** | 0.760 | 0.747 |
| LPIPS (↓) | 0.271 | 0.218 | 0.206 | 0.203 | **0.201** | 0.202 | 0.207 | 0.244 | 0.271 | 0.226 | 0.214 | 0.209 | 0.205 | **0.202** | **0.202** | 0.227 |
| FID (↓) | 37.78 | 32.81 | 30.76 | 29.89 | 29.33 | **28.99** | 29.85 | 37.22 | 37.78 | 35.07 | 33.17 | 32.25 | **29.29** | 29.97 | 29.71 | 34.22 |

Table 4: Step size ablation on the step size for CDDB $c := \rho_i / (1 - \alpha_{i|i+1}^2)$ using 100 test images on SR×4-bicubic task, NFE=100. I$^2$SB [26], Chosen step size.

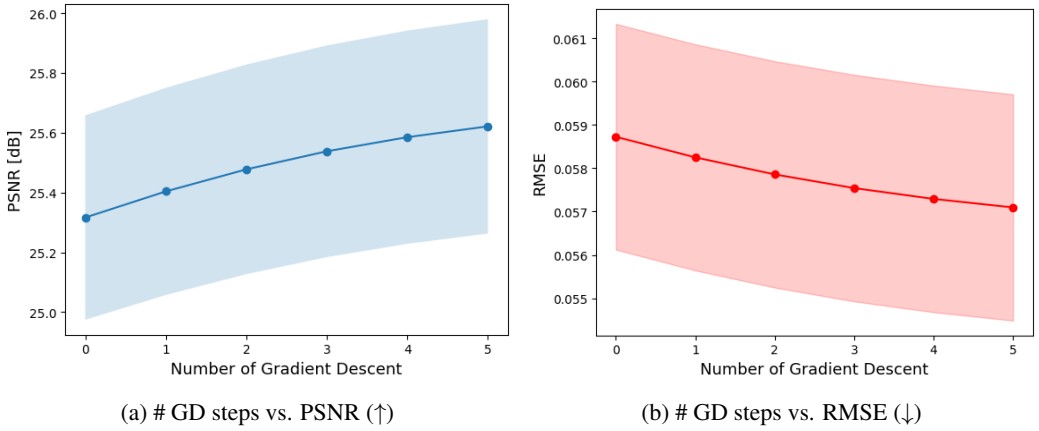

(a) # GD steps vs. PSNR (↑)   (b) # GD steps vs. RMSE (↓)

Figure 6: Effect of gradient descent (GD) steps in CDDB applied to intermediate samples $\hat{x}_{0|i}$. Mean $\pm 0.1$ std indicated as colored region.

**DDS**   We use 100 NFE sampling with $\eta = 0.85$ since we did not observe additional performance gain with larger NFE.

**PnP-ADMM, ADMM-TV**   Following [5], we use the implementation provided in the `scico` library [2]. For PnP-ADMM, we use $\rho = 0.2$, `maxiter`$= 12$ with the DnCNN [42] denoiser. For ADMM-TV, we use as the regularization term $\lambda \|\boldsymbol{D}\boldsymbol{x}_0\|_{2,1}$ with $(\lambda, \rho) = (2.7e - 2, 1.4e - 1)$ for deblurring and $(\lambda, \rho) = (2.7e - 2, 1.0e - 2)$ for SR.

## C.3   Evaluation

Out of the default 10k evaluation images from 256×256 ImageNet [10] used in [26], we use 1k images by interleaved sampling: i.e. Images of index 0, 10, 20, ... are used. When computing the FID score, we use the `pytorch-fid` package and compare the distribution of the ground truth images vs. the reconstructed images, rather than comparing against the training data, following the setting of [5].

# D   Ablation Studies

## D.1   Step size

In Table 4, we summarize the effect of the choice of step size when implementing CDDB/CDDB-deep. Note that for *all* choice of step sizes, the quantitative metrics are superior to the I$^2$SB counterpart, which states that CDDB stably improves the performance. We observe that for both methods, $c = 1.0$ strikes a good balance between the distortion and the perception metrics. For CDDB-deep, taking a slightly larger step size (e.g. $c = 1.5$) improves performance for certain problems, but we keep $c = 1.0$ for unity.

## D.2 Effect of gradients during sampling

In Fig. 6, we investigate the effect of the CDDB gradient steps when applied to $\hat{x}_{0|i}$ (i.e. intermediate denoised estimate) with $i = N//2^6$, 10 NFE for 300 test samples on SR$\times$4-bicubic task. Note that we can apply multiple GD steps to the intermediate denoised sample $\hat{x}_{0|i}$, which is indicated in the $x$-axis of the figures. Even when we apply multiple GD steps, we observe that the metrics constantly get better, establishing the stability of the proposed method.

---

[6]python notation