# OpenReview forum: "Direct Diffusion Bridge using Data Consistency for Inverse Problems"
_NeurIPS.cc/2023/Conference — NeurIPS 2023 poster_

### Official Review · Reviewer_Qo4y · 2023-06-16

**Soundness:** 3 good
**Presentation:** 3 good
**Contribution:** 3 good
**Rating:** 5
**Confidence:** 2

**Summary:**

This paper focuses on the diffusion model-based inversion problem. The paper first analyzes the current works and unifies them with the Direct Diffusion Bridges. Then, the authors point out that the data consistency is ignored by current works and propose CDDB. Experiments show that the proposed CDDB works well on various inverse tasks like inpainting, super-resolution, and deblurring.

**Strengths:**

1. This paper summarized the current work in a systematic way.

2. The differences between this work and previous works are clear.

3. The paper is supported well by theorems and experiments.

**Weaknesses:**

I am not so familiar with the related works and it is hard for me to judge the novelty and contribution of this paper, but the logic of this paper is reasonable, so I can only give borderline accept and I will increase my score if more experts support this paper. I am also wondering whether the work can be applied to the current popular textual inversion problem, since its current form only supports from one noisy image to a clean image.


**Questions:**

Please refer to the weakness part.

**Limitations:**

Some limitations are discusses in the Section 5.

---

> ### Author Rebuttal · Authors · 2023-08-09
>
> We appreciate your honesty and the best efforts to give a constructive feedback. While we are unsure if the method will be applicable to the case of personalization (e.g. textual inversion), please see general comment 3, where we discuss potential applications that are beyond the setting of inverse problems.

---

> > ### Comment · Reviewer_Qo4y · 2023-08-19
> >
> > Thanks for the author's rebuttals. I have read the rebuttal and the reviews from the other reviewers. The technical contribution is OK for this paper, but the application scenario may be a little limited. After consideration, I keep borderline acceptance towards this paper.

---

### Official Review · Reviewer_UzBK · 2023-07-02

**Soundness:** 3 good
**Presentation:** 3 good
**Contribution:** 3 good
**Rating:** 7
**Confidence:** 3

**Summary:**

In this paper, the authors propose Consistent Direct Diffusion Bridge (CDDB) a modification of the Direct Diffusion Bridge (DDB) procedure [1,2] that includes a data consistency term similar to [3,4]. The authors propose two ways to introduce the consistency: the first one forgets about the Jacobian term which is hard to compute in the guidance approximation. The second one (called CDDB-deep) compute the gradient through the observation term. It is more costly but can lead to improved results. The authors briefly justify their method in the spirit of [5] and then showcase the efficiency of their method on several inverse problems (inpainting/deblurring) on ImageNet 256x256.

[1] Liu et al. (2023) -- SB: Image-to-Image Schrödinger Bridge

[2] Delbracio, Milanfar (2023) -- Inversion by Direct Iteration: An Alternative to Denoising Diffusion for Image Restoration

[3] Song et al. (2023) -- Pseudoinverse-Guided Diffusion Models for Inverse Problems

[4] Chung et al. (2023) -- Improving Diffusion Models for Inverse Problems using Manifold Constraints

**Strengths:**

* The paper is very well-written and the explanations are very clear. The presentation of the method is sound and the authors did a good job at recalling the basics on diffusion models and I2SB.

* I really appreciated the discussion on the difference (and ultimately the equivalence) between I2SB [1] and INDI [2]. This discussion is very well presented.

* The experiments are strong. The authors have done a very thorough work investigating several metrics on challenging problems in ImageNet 256x256.

* While one could complain that the novelty of the method is relatively low (basically combining I2SB with a guidance term), I think the experiments are extensive enough to warrant publication.

**Weaknesses:**

* I think this is overall a good paper and the next point is not the central problem tackled by the paper but I think it should be clarified (or at least the related remarks should be tamed). The authors spend a few lines deriving the variance preserving condition (Equation (12)) but not a lot is said afterwards. The authors basically refer to [1] to justify their method "Furthermore, subsequent noising process using deterministic and stochastic noises can then be used to ensure the transition to the correct noisy manifold". It is not clear to me that the results of [1] apply directly in the context of I2SB. Also, the current work is not self-contained as it relies heavily on the work of [1] and in particular the conjugate gradient method is never explained in the manuscript. I think that in the related work or in the theoretical section the authors should put more effort into giving details about what is [1] and how to adapt it to the current context (even though this is quite clear from reading the paper). In the experiments, the authors claim that that "CDDB generally has higher speed and stability, possibly due to guaranteed convergence". In the paper, no convergence results are provided. I don't see anything that would suggest that CDDB has guaranteed convergence. Overall, I think that the theoretical analysis is the weak point of the paper.

* The authors "exclude PiGDM for the baseline comparison in the deblurring problem, as directly leveraging the pseudo-inverse matrix may result in unfair boost in performances". If this is the case then why not include PiGDM and use CDDB-deep which is also preconditioned with the pseudo-inverse? The fact that PiGDM performs (better?) than the proposed method because it makes a smart use of the pseudo-inverse does not seem like a good reason to remove it from the baseline.

[1] Chung et al. (2023) -- Fast Diffusion Sampler for Inverse Problems by Geometric Decomposition

**Questions:**

* Another method that performs transfer tasks is Rectified Flow (albeit in a different context) [1] (or other transport based methods like [2]). Since these methods are not optimised to promote the fidelity to the term $Ax=y$ they should have a higher FID score but they might give better LPIPS, especially Rectified Flow which is deterministic. Can the authors comment on that?

* I find the comment on the link with Schrodinger Bridge to be quite interesting but also quite confusing. In SB problems the coupling is not known beforehand. Hence, I don't really see how one could leverage a relation of the form $Ax=y$.

* is it also possible to use some preconditioning with the pseudo-inverse in CDDB? (since in CDDB-deep "we find that preconditioning with the pseudo inverse as in PiGDM improves performance" it is a natural question.

*Line 87: there is no footnote associated with the superscript 1.

[1] Liu et al. (2023) -- Flow Straight and Fast: Learning to Generate and Transfer Data with Rectified Flow

[2] Shi et al. (2023) -- Diffusion Schrödinger Bridge Matching

**Limitations:**

Adressed.

---

> ### Author Rebuttal · Authors · 2023-08-09
>
> **W1. Weak background for DDS, theory over-claimed**
>
> **A.**
> Thank you for your encouraging comments. In the revised manuscript, we will spend more effort on the discussion of [1]. Moreover, we agree that we cannot guarantee the convergence of CDDB. The statement will be removed from the manuscript.
>
> **W2. Exclusion of methods that leverage pseudo-inverse for deblurring**
>
> **A.**
> We would like to clarify that for all the experiments regarding deblurring, CDDB (Algorithm 1) was used. We refrained from the usage of CDDB-deep (Algorithm 2) to avoid the *inverse crime* as it is based on the perfectly known forward operator and the pseudo-inverse which may unfairly boost the restoration performance for the noiseless deblurring problem. Therefore, the metrics that were reported in Table 2 were obtained through CDDB (Algorithm 1), and hence no pseudo-inverse operation was used. This also holds for the SR experiments, and the only places where we do report the metrics on CDDB-deep are Figure 1 and Table 3, each corresponding to SR and JPEG restoration. In our humble opinion, this gives us the justification not to compare with $\Pi$GDM.
>
> **Q1. Comparison with Rectified Flow, Diffusion Schrödinger Bridge Matching**
>
> **A.**
> Thank you for pointing this out. Indeed, more general transport-based methods such as rectified flow and diffusion schrödinger bridge matching, which do not require paired data from the source and the target measure, but instead match the distributions directly, could induce better perceptual quality. That said, we are a bit confused with the reviewer’s comment, as we usually see lower FID and better LPIS at the same time regardless of the distribution matching or DDB with paired data. We therefore assume that the reviewer intended to mean worse distortion metric (PSNR/SSIM), but better perceptual metric (FID/LPIPS) - but please correct us if we are wrong). Although out of the scope of this work, if these methods were applied for solving inverse problems, CDDB-like consistency regularization would indeed boost the sample quality by correcting the improper deviations. However, we would like to note that while schrödinger bridge  methods are more versatile and can be applied to more general tasks, e.g. image translation, they are generally unstable especially when scaling to high-dimensional data. To the best of our knowledge, there has been not much progress in using general schrödinger bridge  methods for solving inverse problems in imaging.
>
> **Q2. Extension to Schrödinger Bridge Matching**
>
> **A.**
> Please see general comment 3. While there exists no explicit notion of measurement consistency in SB problems, we could still induce a relaxed notion of consistency to constrain the transport path to be consistent with respect to the starting point. For instance, one could leverage a contrastive loss as in [1], or introduce a notion of cycle-consistency by jointly training a network that retrieves a sample from the starting measure.
>
> **Q3. Pre-conditioning for CDDB**
>
> **A.** This is indeed a natural question to ask. Unfortunately, so far we did not observe any performance gain through pre-conditioning for the case of CDDB. This issue may need further investigation in the future.
>
> **References**
>
> 1. Kim, et al. "Unpaired Image-to-Image Translation via Neural Schr\"odinger Bridge." arXiv preprint arXiv:2305.15086 (2023).

---

> > ### Comment · Reviewer_UzBK · 2023-08-12
> > **Answer to rebuttal**
> >
> > I would like to thank the author for the rebuttal. I am satisfied with their answer. I have a few other minor comments and questions.
> >
> > > Having said that, indeed, CMs share important aspects with DDBs, especially considering the inference procedure (i.e. Algorithm 1 of [3]). Since the network is distilled to produce a one-step estimation of the endpoint of PF-ODE trajectory, it can directly produce a clean image [...]
> >
> > (In the reply to the Reviewer 7H1C) I fail to see how this is specific to consistency models? If the network is parameterised to give the `x`-predicition and not the score prediction then the same conclusion can be drawn. Am I missing something specific to consistency models?
> >
> > > However, we would like to note that while schrödinger bridge methods are more versatile and can be applied to more general tasks, e.g. image translation, they are generally unstable especially when scaling to high-dimensional data. To the best of our knowledge, there has been not much progress in using general schrödinger bridge methods for solving inverse problems in imaging.
> >
> > I would like to point out that I2SB actually does compute the Schrodinger bridge if the given coupling is the entropic regularized OT one. In that respect I think the authors could tame their comment. However, I agree with the reviewer that so far the SB framework has received less attention.
> >
> > In the paper the authors claim that I2SB is a "Schrodinger bridge with paired data". This is actually not true. I2SB only solves the Schrodinger bridge problem if and only if the paired coupling is the entropic OT coupling which is usually not the case. While I don't blame the authors for this mistake (I2SB being slightly misleading in that respect), I think it would be extremely valuable to correct the narrative. In particular, I2SB does not solve an OT problem (note that this is not necessarily a problem as for paired setting it is not clear why solving a OT problem would be useful).
> >
> > > That said, we are a bit confused with the reviewer’s comment, as we usually see lower FID and better LPIS at the same time regardless of the distribution matching or DDB with paired data.
> >
> > Sorry if my comment was unclear let me clarify. I was simply stating that Schrodinger bridge methods (which truly solve the OT problem like Rectified flow, DSB or DSBM) are trained to minimize the L2 cost between the samples from the blurry and the samples from the clean distribution, see for example the "straight coupling" property discussed in the Rectified Flow paper. As such one would expect that the similarity between the blurry and clean samples would be greater. I was merely asking what are the thoughts of the authors on this point. I am happy to clarify further.

---

> > > ### Author Response · Authors · 2023-08-13
> > >
> > > We are glad that you find our rebuttal satisfactory. For your additional comments, see below.
> > >
> > > > (In the reply to the Reviewer 7H1C) I fail to see how this is specific to consistency models? If the network is parameterised to give the ```x```-predicition and not the score prediction then the same conclusion can be drawn. Am I missing something specific to consistency models?
> > >
> > > Indeed, both consistency models and DDBs are capable of producing $x$-predictions at every timestep $t$. However, there are two important distinctions. 1) DDBs produce x-predictions given the degraded measurements (or some convex combination between $x_0$ and $y$), while CMs along with other DIS produce $x$-predictions given the Gaussian noise (or some $x_t$ along the trajectory).
> > > 2) Although when converged to optimality, CMs will be able to produce the endpoints $x_0$ of the diffusion trajectory, the main difference between CMs from other diffusion models is the introduction of the distillation processes between the time points which results in a significant reduction of the sampling steps.
> > >
> > > > I would like to point out that I2SB actually does compute the Schrodinger bridge if the given coupling is the entropic regularized OT one. In that respect I think the authors could tame their comment. However, I agree with the reviewer that so far the SB framework has received less attention.
> > >
> > > Thank you for your detailed comments! We agree that I2SB is a Schrödinger bridge under circumstances, and in this regard, we can definitely see that SBs can be stable and scalable, even in the context of inverse problems. Our former sentence intended to target SBs that do not require paired data from two domains. We would like to take back our claim that “To the best of our knowledge, there has been not much progress in using general Schrödinger bridge methods for solving inverse problems in imaging.”.
> > >
> > > > In the paper the authors claim that I2SB is a " Schrödinger bridge with paired data". This is actually not true. I2SB only solves the Schrodinger bridge problem if and only if the paired coupling is the entropic OT coupling which is usually not the case. While I don't blame the authors for this mistake (I2SB being slightly misleading in that respect), I think it would be extremely valuable to correct the narrative. In particular, I2SB does not solve an OT problem (note that this is not necessarily a problem as for paired setting it is not clear why solving a OT problem would be useful).
> > >
> > > Thank you for pointing this out. We agree that **"Schrodinger bridge with paired data"** is not the best one-line characterization of I2SB. We will modify the statement to be **“Paired diffusion motivated by Schrödinger bridge”**.
> > >
> > > > Sorry if my comment was unclear let me clarify. I was simply stating that Schrodinger bridge methods (which truly solve the OT problem like Rectified flow, DSB or DSBM) are trained to minimize the L2 cost between the samples from the blurry and the samples from the clean distribution, see for example the "straight coupling" property discussed in the Rectified Flow paper. As such one would expect that the similarity between the blurry and clean samples would be greater. I was merely asking what are the thoughts of the authors on this point. I am happy to clarify further.
> > >
> > > Thank you for the clarification. We believe that this might be the case if Rectified Flows were to be modified to be trained with paired matching data. This is definitely an interesting venue of research, and further investigation would be needed.

---

### Official Review · Reviewer_TLjQ · 2023-07-07

**Soundness:** 3 good
**Presentation:** 3 good
**Contribution:** 3 good
**Rating:** 6
**Confidence:** 4

**Summary:**

This paper studies the existing works about Direct Diffusion Bridges (DDB) with a unified scheme and limitations, and proposes a modified inference procedure that imposes data consistency without the need for fine-tuning, called data Consistent DDB (CDDB), as a new diffusion model-based inverse problem solvers.


**Strengths:**

+ The paper is well written and organized. Especially, the writing of the background section clearly describes and sorts out several important relevant works in a unified perspective.

+ This paper solves an important problem to study the data consistency problem with diffusion models, which can be used to solve inverse problems generally.

+ The experiments performed on natural images are comprehensive, and results show satisfying reconstruction quality, with comparison methods in both DIS and DDB methods.


**Weaknesses:**

- The paper studies a series of DDB methods which match the clean data/image distribution with measurements distribution, with the key contribution to add the data consistency term into the inference sampling steps. Actually, this kind of data consistency term has been introduced in quite a few previous works about diffusion model-based inverse problem solving, so called DIS/DDS methods in this paper. Although this paper claims the proposed approach is a generalization of the DDS method, after reading the whole manuscript, one essential question is still not very clear why DDB methods should outperform DIS/DDS methods, if they share a similar scheme from some perspective and both consider data consistency constraints during the sampling steps.

- As mentioned above, the idea of updating the clean part for data consistency from DDIM formulation has been introduced in some previous works, such as DDS paper [5] and DDNM paper [39]. Although the paper claims that the proposed approach is a generalization of method in [5], the comparison with these two similar papers is missing in the experiments.

- Besides, for the proposed DDB methods, it seems to require the paired data for training the distribution matching, different from the series of DIS methods which generally are trained without paired data. Although the paper discuss the relevance with supervised learning frameworks, the proposed method may need to be also compared with other supervised methods or conditional diffusion methods, which are missing in the current experiments and results. Besides, in the Discussion section, the paper claims the CDDB is flexible and does not have to pre-determine the number of forward passes or modify the training algorithm. This flexibility is also carried with all the DIS methods generally which are used to solve inverse problems based on trained diffusion models, but not the unique characteristics from DDB methods, which may need to be clarified clearly in the paper.

- In the proposed CDDB (deep) method, it relies on estimating a pseudo-inverse to preserve data consistency in each sampling step. But for solving non-linear inverse problems, it rarely uses a pseudo-inverse formulation, which may not be straightforward to obtain pseudo-inverse for most nonlinear problems.

- As mentioned in the Discussion, in order to show the advantage of DDB methods to match two distributions with data consistency, a great example is image translation problem, which is kind of surprising not included in the scope of this paper. But meanwhile, for image translation problem, since the forward matrix A is not explicit, is it still possible to use the proposed CDDB method to preserve the data consistency in the image translation problem?


**Questions:**

Please see the weaknesses for specific questions.

Minor comments:
- In Table 2, the best results are not bolded correctly in the PSNR of “pool” images as mentioned in the caption?


**Limitations:**

The authors discuss the limitations and societal impact of the proposed method at the end of the paper.

---

> ### Author Rebuttal · Authors · 2023-08-09
>
> **W1. Why would DDB outperform DIS?**
>
> **A.**
> The main reason that DDB often outperforms DIS (DDS falls into this category) is that DDBs are trained *specifically* for a given task with paired datasets, rather than being a *general* solver. Moreover, it learns to directly start the diffusion process from the corrupted measurement, rather than from Gaussian noise. These two factors both compromise the ability of DDB to be a general solver but enhance the ability as a specific image restoration solver, especially given the same computational budget. We believe that this is an inevitable trade-off.
>
> **W2, 3. Further comparisons**
>
> **A.** Thank you for pointing this out. Please see below for the additional comparison against DDNM and DDS. Further, we did our best to include as many supervised learning and conditional diffusion baselines as additional comparison methods under the limited time span. We plan to add an exhaustive list of comparisons in the future modified version.
>
> We will clarify that the flexibility holds not only for DDB methods but also for DIS methods. However, it is worth noting that for DIS, we do not generally see a smooth Pareto-frontier trade-off between perception and trade-off, as in the very low NFE regime of < 20, the methods usually fail catastrophically.
>
> |        | SR x4     |           |           |           |           |           |           |           |
> |--------|-----------|-----------|-----------|-----------|-----------|-----------|-----------|-----------|
> |        | Bicubic   |           |           |           | Pool      |           |           |           |
> | Method | PSNR      | SSIM      | LPIPS     | FID       | PSNR      | SSIM      | LPIPS     | FID       |
> | CDDB   | **26.41** | **0.860** | **0.198** | **19.88** | **26.36** | **0.855** | **0.184** | **17.79** |
> | I2SB   | 25.22     | 0.802     | 0.260     | 24.13     | 25.08     | 0.800     | 0.258     | 23.53     |
> | DDNM   | **26.41** | 0.801     | 0.230     | 38.63     | 26.04     | 0.792     | 0.218     | 33.15     |
> | DDS    | **26.41** | 0.801     | 0.230     | 38.64     | 26.04     | 0.792     | 0.218     | 33.15     |
> | ESRGAN | 25.08     | 0.792     | 0.244     | 26.38     | 24.90     | 0.803     | 0.203     | 29.38     |
> | SR3    | 24.83     | 0.769     | 0.229     | 23.46     | -         | -         | -         | -         |
>
> **W4. Reliance on pseudo-inverse? Non-linear inverse problems?**
>
> **A.**
> Good point. While the experiments in the current manuscript focus on linear and semi-linear (JPEG restoration) inverse problems, it could be hard to obtain an estimate of the pseudo-inverse for more complex non-linear inverse problems. In such cases, however, we could easily use a DPS-like gradient step without preconditioning, with the reconstruction quality that does not differ too much from CDDB-deep (L191). We show the results with and without the preconditioning here.
>
> |                                | JPEG   |       |       |       |       |       |       |       |
> |--------------------------------|---------|-------|-------|-------|-------|-------|-------|-------|
> |                                | bicubic |       |       |       | pool  |       |       |       |
> | Method                         | PSNR    | SSIM  | LPIPS | FID   | PSNR  | SSIM  | LPIPS | FID   |
> | CDDB-deep (preconditioned)     | 26.81   | 0.876 | 0.177 | 18.25 | 26.70 | 0.865 | 0.188 | 17.81 |
> | CDDB-deep (not preconditioned) | 26.73   | 0.872 | 0.185 | 18.90 | 26.59 | 0.860 | 0.189 | 18.22 |
>
> **W5. Extension to image translation**
>
> **A.** Please see general comment 3. We agree that it would be very interesting to scale CDDB to image translation problems. However, DDB-type algorithms are not generally suitable for image translation where the exact match between the two domain images are difficult to obtain (eg. horse-to-zebra, Monet-to-Van Gogh). Even if this was the case as the reviewer mentions, we do not have an explicit forward matrix $A$ for image translation problems and hence would have to resort to an approximated pseudo-forward operator. For instance, if we could jointly train another network that is trained to recover back the starting signal as in cycleGANs, we would be able to use the neural network as our forward operator and use an algorithm similar to CDDB-deep to impose consistency on the starting image.
>
> **Minor Comment**
>
> **A.** Fixed.

---

### Official Review · Reviewer_7H1C · 2023-08-01

**Soundness:** 3 good
**Presentation:** 2 fair
**Contribution:** 2 fair
**Rating:** 5
**Confidence:** 3

**Summary:**

The paper introduces a unified view of several existing diffusion-model-based methods for solving the inverse problem (i.e., DPS and $\Pi$GDM), and proposes a new approach for this problem called Direct Diffusion Bridge (DDB) which is derived from the DDPM ancestral sampling, and to some extent, links to the Image-to-Image Schrodinger bridge (I$^2$SB). The main idea behind DDB is using a trained neural network $G_{\theta^*}$ to compute a denoised image $\hat{x}\_{0|t}$, which is an approximation of $x_0$, from $x_t$. The paper also proposes an improved version of DDB called Consistent DDB (CDDB), which according to the authors, is more beneficial for reconstruction (of $x_0$).

**Strengths:**

- The unified view between DPS and $\Pi$GDM is interesting though I think it’s not hard to figure out as both methods share the same motivation.
- Experimental results seem to support the proposed method there are improvements in both perception and signal-to-noise ratio compared to previous works. However, the fairness in settings of different methods should be clarified in the paper.

**Weaknesses:**

1) The presentation in the paper makes the motivation for the proposed method “Direct Diffusion Bridge” (DDB) unclear. From my view, neither the unified view between DPS and $\Pi$GDM (Section 2.2) nor the I$^2$SB model (Section 3.1) motivates the design of the denoised image $\hat{x}_{0|t}$ in the paper. In fact, it seems like in Section 3.1 the authors try to link the posterior of $x_t$ in the I$^2$SB model (Eq. 7) with the posterior of $x_s$ (s < t) in the DDPM (Eq. 10), which, I guess, leads to the name “Direct Diffusion Bridge”.

2) I think the main contribution *in terms of technique* in this paper is a new way to compute $\hat{x}\_0$ (line 2 in Algorithm). Other parts of Algorithm 1 (lines 3 -> 7) were already proposed in previous works (e.g., DPS, DDS). However, the trained network $G\_{\theta^*}$, which plays an important part in the algorithm, is not described in detail in the paper. It is only mentioned briefly at lines 120 -> 121. Besides, no ablation study was conducted to validate the use of $G\_{\theta^*}$ with other methods for computing $\hat{x}\_0$ from $x\_t$ (e.g., Tweedie’s formula).

3) The authors don’t mention how $G\_{\theta^*}$ can handle $x\_t$ with different time steps $t$ in the paper. Does it take both $x_t$ and $t$ as input? In fact, from my view, $G\_{\theta^*}$ is very similar to the network in the paper Consistency Models [1]. However, I don’t see the authors discuss Consistency Models in their paper. Since Consistency Models can be directly used for DIS (a lot of experiments on inverse sampling can be found in [1]), I am keen to see the empirical comparison between this method and Consistency Models.

4) Please correct me if I am wrong but I cannot find the NFE of the proposed method in the main paper or the supplementary material. Therefore, it is hard to fairly compare the performance of the proposed method with other baselines.

[1] Consistency Models, Song et al., ICML 2023

**Questions:**

1) What are the architecture and configurations of the residual network $G\_{\theta^*}$? I couldn’t find any detail about $G\_{\theta^*}$ in the main paper or the supplementary material. Does it depend on time $t$ besides $x_t$

2) What are the technical differences between the method in this paper and the use of Consistency Models for the inverse sampling problem?

3) In Algorithm 1, is $x\_1$ $y$ or Gaussian random noise? Since I$^2$SB can take the corrupted image $y$ as input, do the authors think their method can do so?

4) In Eq 15, $\mathbb{E}[x_t]$ should be $\mathbb{E}[x_0|x_t]$.

5) The main difference between Algorithm 1 and Algorithm 2 is the computation of $g$. In Alg. 1, $g$ is computed as $\nabla_{\hat{x}\_{0|t}} \log p(y|\hat{x}\_{0|t})$ while in Alg. 2, $g$ is computed as $\nabla_{x_t} \log p(y|\hat{x}\_{0|t})$. In my view, Alg. 2 is *more theoretically correct* than Alg. 1 rather than just being more beneficial for reconstruction as mentioned in the paper in lines (181 -> 183). Thus, I hope the authors could make this point clearer in their paper rather than just mentioning CDDB as an alternative for DDB when the inverse problem is non-linear. Even when the inverse problem is linear, we still need the term $\frac{\partial \hat{x}\_{0|t}}{\partial x_t}$ which is the Jacobian in Eqs. 5, 6.

6) At line 181, the authors should be clear about the “U-Net Jacobians”. Is the U-Net here the noise network $\epsilon$ of the diffusion model or the network $G\_{\theta}$? I guess it should be $G\_{\theta}$ since we are considering $\frac{\partial \hat{x}\_{0|t}}{\partial x_{t}}$. The authors give me the impression that they are trying to hide details about $G_{\theta}$, which in fact plays a critical role in the paper.

**Limitations:**

Please refer to my questions above

---

> ### Author Rebuttal · Authors · 2023-08-09
>
> After reading the comments, we would like to respectfully stress that there seems to be a **clear misunderstanding** of the paper. 1) Our work aims for the unified view of DDB, not DIS; 2) We do not introduce a new way to compute $\hat{x}_0$, which is directly estimated from the neural network. Only the inference process changes; 3) Information about the network architecture, NFE, etc. are already clearly stated in the paper, which by no means do we attempt to hide. For detailed responses on each of the points, please see below.
>
> **W1. Motivation of the term “Direct Diffusion Bridge”?**
>
> **A.**  We emphasize that the aim of the paper was **not** to propose a unified view between DPS and PGDM, **nor** to link the posterior of I2SB with the posterior of DDPM.
> In fact, the first aim of this paper was to unify the seemingly different approaches---I2SB and InDI---under a common framework. Note that different from DIS approaches (e.g. DPS, PGDM), these methods are trained in a paired fashion to directly invert the degraded image. This is most clearly seen in the case of t=1, where it would correspond to the simple supervised learning setting, hence the term **Direct**. Furthermore, these strategies train a neural network with multiple levels of degradation such that at test time, one can utilize an ancestral sampling procedure used in diffusion models for iterative refinement, hence the name **Diffusion**.
>
> **W2. How to compute $\hat{x}_0$? Ablation studies?**
>
> **A.**
> We are not introducing a new way to compute $\hat{x}_0$. In fact, $\hat{x}_0$ is a direct output of the neural network up to a pre-defined multiplicative and additive constant.
> This can be interpreted as the minimum mean squared error (MMSE) estimate $\mathbb{E}[x_0|x_t]$ up to parametrization/optimization error because the neural network was trained to minimize the l2 distance against the target $x_0$ given $x_t$. Note that $x_t$ here is different from the usual notion of $x_t$ in the diffusion model literature, where it is a Gaussian noised version of $x_0$. Rather, it is given by (8). Hence, we cannot use Tweedie’s formula directly as 1) retrieving $x_0$ from $x_t$ is not denoising, and 2) the trained neural network is not a score function. However, note that Tweedie’s formula is just a way of computing the posterior mean for the case of Gaussian noisy images. In this regard, the method of producing $x_0$ can be thought of as analogous to Tweedie’s formula for general degradations arising in direct diffusion bridges.
>
> **W3, Q1, Q2. Net. Arch., Comparison with CM[3]?**
>
> **A.**
> We take the neural net $G_\theta$ directly from I2SB without retraining (L194-195), and hence the model architecture follows that of ADM [4], which takes in both $x_t$ and $t$ as input.
>
> In Consistency Models (CM)[3], $x_t$ is a Gaussian noisy version of $x_0$ as in diffusion models, and not as in DDBs. In this regard, solving inverse problems using Algorithm 4 of [3] falls into the category of DIS, and is especially similar to [5] and [6]. Note that our technical contribution is to take an existing trained DDB, and modify the inference procedure such that it boosts performance while being more faithful to the measurement. Thus, including a comparison against CMs would be adding yet another DIS to the current list of DIS methods used for comparison (DPS, PGDM, DDRM, DDNM), which, in our humble opinion, could be beneficial, not necessary.
>
> Having said that, indeed, CMs share important aspects with DDBs, especially considering the inference procedure (i.e. Algorithm 1 of [3]). Since the network is distilled to produce a one-step estimation of the endpoint of PF-ODE trajectory, it can directly produce a clean image $\hat{x}_0$ from $x_t$. It would be interesting to discuss the similarities and differences between the proposed method and CMs, which will be included in the modified discussion. In the current official repo of CMs, ImageNet 256x256 checkpoint is missing and we would have to train the model from scratch, which would be infeasible within a week considering the limited computational resources. However, we do plan to include this comparison after the rebuttal period and include them in the revised manuscript.
>
> **W4. NFE of CDDB**
>
> **A.** Please see L210-211. JPEG restoration: 100 NFE, others: 1000 NFE.
>
> **Q3.**
>
> **A.** Please see L111. $x_1=y$, the same way as in I2SB.
>
> **Q4.**
>
> **A.** Fixed.
>
> **Q5, Q6. Correctness of incorporating U-Net Jacobian**
>
> **A.** We agree that $\nabla_{x_t}$ is more exact. We will make this clearer in the revised manuscript. However, we would like to note that in practice, due to the U-Net Jacobian being unstable, it is often beneficial to skip this, consistent with observations made in [7,8].
>
> We respectfully disagree with your comment that we are hiding details about the network. U-Net Jacobian means the Jacobian of $G_\theta$. It is very explicit that we take the models directly from I2SB, and give discussions on the model architecture (L194-197). Among the lines, we also mention that the model architecture stems from ADM, which is U-Net based, hence the name U-Net Jacobian, similar to how the term is used in diffusion model literature.
>
> **References**
> 1. Guan-Horng et al. "Image-to-image Schrödinger Bridge" ICML (2023).
> 2. Delbracio and Milanfar. "Inversion by direct iteration: An alternative to denoising diffusion for image restoration." TMLR (2023).
> 3. Yang et al. “Consistency models." ICML (2023).
> 4. Prafulla and Nichol. “Diffusion models beat gans on image synthesis.” NeurIPS (2021)
> 5. Yang et al. “Score-based generative modeling through stochastic differential equations.” ICLR (2021).
> 6. Wang et al. “Zero-shot image restoration using denoising diffusion null-space model.” ICLR (2023).
> 7. Poole et al. “Dreamfusion: Text-to-3d using 2d diffusion.” ICLR (2023).
> 8. Chung et al. “Fast Diffusion Sampler for Inverse Problems by Geometric Decomposition.” arXiv (2023).

---

> > ### Comment · Reviewer_7H1C · 2023-08-19
> > **Comment on the authors' rebuttal**
> >
> > I would like to thank the authors for their rebuttal. It has addressed most of my concerns. The authors' answers clarified my misunderstanding of their paper at first sight. I thought their work was a variant of DIS. After the authors' rebuttal, I can see that their work belongs to the class of bridging methods. The reason why the authors didn't give many details about $G\_{\theta}$ and other training settings is now clear. It is because their proposed method CDDB uses exactly the same training procedure and settings of $I^2SB$ and is only different from $I^2SB$ in the integration of the gradient $g$ (Algorithms 1 and 2) during the generation stage. In fact, using $g$ provides more signals to adjust  $x\_{t-1}$ but also has a limitation that it assumes $y=Ax$ and may not be applicable to general settings where $y = f(x)$ with $f$ is an arbitrary function.
> >
> > I decide to raise my score to 5.

---

### Author Rebuttal · Authors · 2023-08-09

We would like to thank the reviewers for their constructive and thorough reviews. We are encouraged that the reviewers think that our paper “provides an interesting unified view, with support of experimental results” (7H1C), is “well written, organized, with comprehensive experiments” (TLjQ), and is “very well presented, clear, with strong experiments” (UzBK).

We have summarized some of the major concerns that were raised by the reviewers below. Point-to-point responses were also included as a reply to each reviewer. We have also added a PDF file for additional experimental results.

**1. Clarification of the contribution, the reason for DDB being better than DIS**

We would like to clarify that the main contribution of the paper is two-fold: 1) Unification of seemingly different theories under the framework of direct diffusion bridges (DDB), and devising a new sampling algorithm that additionally imposes data consistency on DDB to yield better reconstruction results (CDDB). This direction is orthogonal and complementary to the recent endeavors to improve existing Diffusion Model-based Inverse problem Solver (DIS). A major difference, and also the reason why DDB often outperforms DIS is because DDBs are trained for specific inverse problems using paired datasets rather than being a universal solver for all tasks. As the inference distribution is pre-defined and tuned for each task, DDBs offer enhanced quality and stability.

**2. More extensive comparisons including DIS (PIGDM, DDNM, DDS), conditional diffusion-based methods (Palette), and supervised learning methods.**

While we offered quite extensive set of comparison methods in the original submission, some of the reviewers pointed out several important works that are worth comparing against. Per the requests, we include the modified tables that now include additional methods.

**3. Discussion on how CDDB can be extended to more general cases, e.g. image translation**

For general translation tasks where there is no coupling between two specific data points (e.g. the general case of Schrödinger bridge), we cannot easily constrain the inference path to follow the consistency $y = Ax$. Even in such cases, we can generalize the notion of consistency, and impose the inference path so that one can regularize the resulting image to be similar to the starting image. One concrete example would be the use of contrastive loss [1], but one could use other strategies such as cycle consistency.


**References**

[1] Kim, Beomsu, et al. "Unpaired Image-to-Image Translation via Neural Schrödinger Bridge." arXiv preprint arXiv:2305.15086 (2023).

---

### Decision · Program_Chairs · 2023-09-21

**Decision:**

Accept (poster)

**Comment:**

This paper presents a modification to direct diffusion bridge to enforce data consistency (following recent works on consistency for acceleration). The paper introduces two variations (one faster and one better reconstruction results) and has a detailed comparison to multiple recent papers. This is a very competitive space and the authors did a great job in evaluating their method and showing very good results. The theoretical claims were over-stated but the authors were willing to revise their manuscript based on valid feedback on this limitation.